# Inhibitory Effect on Nitric Oxide Release in LPS-Stimulated Macrophages and Free Radical Scavenging Activity of *Croton Linearis* Jacq. Leaves

**DOI:** 10.3390/antiox11101915

**Published:** 2022-09-27

**Authors:** Jesús García Díaz, Rosalia González Fernández, Julio César Escalona Arranz, Gabriel Llauradó Maury, Daniel Méndez Rodríguez, Linda De Vooght, Enrique Molina, Emmy Tuenter, Luc Pieters, Paul Cos

**Affiliations:** 1Pharmacy Department, Faculty of Natural and Exact Sciences, Universidad de Oriente, Santiago de Cuba 90500, Cuba; 2Toxicology and Biomedicine Centre (TOXIMED), Universidad de Ciencias Médicas, Autopista Nacional Km 1½ s/n, Santiago de Cuba 90400, Cuba; 3Center of Studies for Industrial Biotechnology (CEBI), Universidad de Oriente, Santiago de Cuba 90500, Cuba; 4Chemistry Department, Faculty of Applied Sciences, Universidad de Camagüey, Carretera de Circunvalación Km 512, Camagüey 74650, Cuba; 5Laboratory of Microbiology, Parasitology and Hygiene (LMPH), Faculty of Pharmaceutical, Biomedical and Veterinary Sciences, University of Antwerp, 2610 Antwerp, Belgium; 6Natural Products & Food Research and Analysis (NatuRA), Department of Pharmaceutical Sciences, Faculty of Pharmaceutical, Biomedical and Veterinary Sciences, University of Antwerp, 2610 Antwerp, Belgium

**Keywords:** *Croton linearis*, nitric oxide release, alkaloids, flavonoids, antioxidants

## Abstract

Oxidative stress is an important component of many diseases including cancer, along with inflammatory and neurodegenerative processes. Natural antioxidants have emerged as promising substances to protect the human body against reactive oxygen and nitrogen species. The present study evaluates the inhibition of nitric oxide (NO) production in LPS-stimulated RAW 264.7 murine macrophages and the free radical scavenging activity of *Croton linearis* Jacq. leaves. UPLC-QTOF-MS analysis identified 18 compounds: nine alkaloids with a morphinane, benzylisoquinoline or aporphine nucleus, and nine *O*-glycosylated-flavonoids with quercetin, kaempferol and isorhamnetin as the aglycones. The crude extract (IC_50_ 21.59 µg/mL) and the *n*-hexane fraction (IC_50_ 4.88 µg/mL) significantly reduced the NO production in LPS-stimulated macrophages but with relatively high cytotoxicity (CC_50_ 75.30 and CC_50_ 70.12 µg/mL, respectively), while the ethyl acetate fraction also showed good activity (IC_50_ 40.03 µg/mL) without affecting the RAW 264.7 cell viability. On the other hand, the crude extract, as well as the dichloromethane and ethyl acetate fractions, showed better DPPH and ABTS free radical scavenging activities. Considering the chemical composition and the activity observed for *Croton linearis* leaves, they may be considered a good source of antioxidants to combat oxidative damage-related diseases.

## 1. Introduction

Oxidative stress has emerged as one of the important factors in the pathogenesis of different diseases such as cancer and diabetes, autoimmune, cardiovascular and neurodegenerative diseases, inflammatory processes, etc. [1,2]. For this reason, interest in the use of natural antioxidants to protect the human body against diseases mediated by reactive oxygen species (ROS), reactive nitrogen species (RNS) and free radicals has increased [3,4,5].

Plants are susceptible to environmental stress and have developed numerous defense systems, resulting in the formation of potent antioxidants [6]. In fact, it is well-accepted in the literature that plants are a natural reservoir of antioxidant bioactive compounds [7]. Among the compounds that are produced by plants, the phenolic compounds (characterized by one or more hydroxyl groups bound to one or more aromatic rings) such as phenolic acids, flavonoids, anthocyanidins and tannins are the major contributors to the antioxidant activity of plant extracts [8,9]. These compounds act by various mechanisms, including binding metal ions, scavenging reactive oxygen and nitrogen species, converting hydroperoxides to non-radical species, absorbing UV radiation and deactivating singlet oxygen [2].

*Croton linearis* Jacq. (Euphorbiaceae) is a small shrub endemic to the Bahamas, Florida and the West Indies, growing mostly in rather dry environments and on rocky land near the coast, in sandy and gravelly soils and open areas [10]. In Cuba, it is a native plant distributed along the southern coast of the eastern region, where the coastal and pre-coastal xeromorphic shrub predominates [11,12]. Ethnopharmacological information refers to its use mainly to relieve pains related to menstrual periods, childbirth and rheumatism, and to treat fever and colds [11,13]. A few studies for this species have shown the presence of isoquinoline alkaloids as the main compounds [14,15], while a diterpene [16], two flavonoids [15,17] and volatile compounds [18] have also been reported. In addition, studies have revealed its antiparasitic potential against *Leishmania* sp. [18], *Trypanosoma* sp. [15] and *Plasmodium* sp. [19], as well as larvicidal activity [20]. According to this phytochemical and pharmacological evidence and considering the dry and harsh environmental conditions of its habitat, it was recently mentioned as one of the medicinal plants growing in the Siboney-Juticí Ecological Reserve (located in the southeast part of Cuba) of interest for its possible but not yet explored antioxidant potential [21].

Considering the absence of investigations regarding the antioxidant properties of *C. linearis* Jacq. leaves, the aim of this study was to evaluate the extract and its fractions for radical-scavenging activity against 2,2-diphenyl-1-picrylhydrazyl (DPPH) and 2,20-azino-bis-(3-ethylbenzothiazoline-6-sulfonic acid) (ABTS), and the nitric oxide (NO) inhibitory activity in LPS-activated RAW 264.7 murine macrophages. ultra-performance liquid chromatography-quadrupole time of flight-mass spectrometry (UPLC-QTOF-MS) was used for the tentative identification of potentially bioactive compounds present in the most active fractions.

## 2. Materials and Methods

### 2.1. Material and Reagents

All solvents, including *n*-hexane, dichloromethane, ethyl acetate, *n*-butanol and dimethyl sulfoxide (DMSO), and ethanol (Emsure grade) were purchased from Acros Organics (Geel, Belgium) or Fisher Scientific (Leicestershire, UK) and were analytical grade. Lipopolysaccharide (LPS) from *Escherichia coli* (0128:B12), tamoxifen, resazurin sodium salt (7-hydroxy-3H-phenoxazin-3-one-10-oxide), 1,1-diphenyl-2-picryl-hydrazyl (DPPH) and 2,2-azino-bis-3-ethylben-zothiazoline-6-sulfonic acid (ABTS) were purchased from Sigma-Aldrich (St. Louis, MO, USA). Dulbecco’s modified Eagle’s medium (DMEM) was sourced from Gibco (New York, NY, USA). A Griess reagent kit, L-NAME (*N*5-(imino(nitroamino)methyl)-L-ornithine methyl ester monohydrochloride) and ascorbic acid (99%,) were purchased from Molecular Probes (Eugene, OR, USA), Alexis Biochemicals (New York, NY, USA) and Fluka (Germany), respectively. UPLC-grade solvents, such as acetonitrile (ACN) and formic acid, were purchased from Biosolve (Valkenswaard, The Netherlands). Water was dispensed by a Milli-Q system from Millipore (Bedford, MA, USA) and was passed through a 0.22 μM-membrane filter before usage.

### 2.2. Samples

*C. linearis* samples, before processing, were collected in the Siboney-Juticí Ecological Reserve, Santiago de Cuba, Cuba, following the guidance from previous studies’ authors [15,19]. After processing, the samples consisted of ethanol crude extract (CEL) and four fractions: *n*-hexane (HF), dichloromethane (DCM), ethyl acetate (EAF) and *n*-butanol (BF). The crude extract (CEL) was obtained by percolation with ethanol 95%, while its fractions (HF, DCM, EAF and BF) were obtained by liquid-liquid fractionation. The extract and fractions were vacuum concentrated until dry at 40 °C (with a Kirka–Werke rotary evaporator) and then stored at −20 °C for further analyses.

### 2.3. Free Radical Scavenging Activity

The antioxidant activity of the crude extract and its fractions was determined using the stable radical DPPH (2,2-diphenyl-1-picrylhydrazyl) and ABTS (2,20-azino-bis (3-ethylbenzothiazoline-6-sulfonic acid). Stock solutions were prepared by dissolving all samples in absolute ethanol at 1 mg/mL. Afterward, the samples were serially diluted to concentrations ranging from 500 to 31.125 μg/mL.

DPPH assay: This was carried out according to previous laboratory protocols [22]. Briefly, 1.5 mL of DPPH solution (0.1 mM in ethanol) was mixed with 0.25 mL of the samples (CLF, HF, DCM, EAF and BF) at different concentrations, shaken and incubated in the dark at room temperature for 30 min. After that, the absorbance was measured against a blank (ethanol) at 517 nm using a spectrophotometer (T60 UV-Visible Spectrophotometer/PG-instruments, UK). The percentage of scavenging activity was calculated using the equation: % scavenging activity = [(A0 − As)/A0] × 100, where A0 is the absorbance of the control (DPPH or ABTS solution without sample), and As is the absorbance of the tested sample (DPPH or ABTS plus sample).

ABTS assay: The ABTS radical scavenging capacity of the extract and its fractions was also measured following previous laboratory protocols [23]. The ABTS radical was produced by incubating a solution of ABTS (7 mM) with a solution of potassium persulfate (2.45 mM) at room temperature for 12 h. The ABTS solution (3 mL) was mixed with the samples (1 mL) at different concentrations, and the absorbance was measured after 90 min at 734 nm using a spectrophotometer (T60 UV-Visible Spectrophotometer//PG-instruments, UK). A solution of 50 μL of distilled water and 3 mL of diluted ABTS was used as the absorbance blank. The ABTS scavenging activity was calculated using the same equation as above.

In both assays, the mean inhibitory concentration (IC_50_) values of the extract and fractions were determined using a non-linear regression curve. Ascorbic acid was used as the positive control (from 31.125 to 500 μg/mL). All the experiments were done in triplicate and the results are presented as the mean ± standard deviation.

### 2.4. Cell Viability Assay

The effect of the extract and its fractions on the RAW 264.7 cell viability was determined by a resazurin dye reduction assay [24]. Briefly, 200 µL of cell suspension (5 × 10^5^ cells/mL) was added to a 96-well plate and incubated in the same conditions mentioned above. After 24 h, the cells were washed twice with 200 µL of DPBS, then fresh DMEM without fetal calf serum (FCS) was added. The samples at different concentrations (8–256 µg/mL) were added to each well and incubated at 37 °C, with 5% CO_2_. Then, 50 µL of resazurin solution (2.2 µg/mL) was added, and the fluorescence was measured after 4 h (λ_excitation_ 550 nm, λ_emission_ 590 nm) using a TECAN GENios microplate reader (Männedorf, Switzerland). Two independent experiments were performed, and the samples were tested in triplicate. The results were expressed as the percentage of reduction in cell growth/viability compared to control wells, and the IC_50_ was determined. Tamoxifen was included as a positive control (at 2–64 µg/mL).

### 2.5. Nitrite Determination

Nitric oxide (NO) production by RAW264.7 macrophages from ATCC (American Type Tissue Culture Collection, USA) was determined by measuring the accumulation of nitrite, an indicator of NO, in the supernatant after 24 h of lipopolysaccharide (LPS) treatment with or without the samples (**CEL, HF, DCM, EAF** and **BF**) using the Griess reagent [24]. The RAW 264.7 cells (5 × 10^5^ cells/mL in a 96-well plate) were activated by incubation in a medium containing 100 ng/mL of lipopolysaccharide (LPS) alone (control), and treated simultaneously with different concentrations of the samples dissolved in DMSO and further diluted in culture medium (from 8 to 64 ug/mL). The concentration of DMSO in the experiment did not exceed 0.5%. After 24 h of incubation, 150 µL of culture supernatant was collected and transferred to a 96-well plate filled with 130 µL of demineralized water. After adding 20 µL of Griess reagent, samples were incubated for 30 min while protected from light, and the absorbance was measured at 540 nm using a TECAN GENios microplate reader (TECAN Group Ltd., Männedorf, Switzerland). A standard calibration curve was set up by diluting the nitrite standard solution of the kit. The percentage of NO inhibition was calculated based on the ability of each sample to inhibit NO production by RAW 264.7 macrophages compared with the control (cells treated with LPS without samples). L-NAME (100 µM) was used as a reference control to suppress NO production in LPS-stimulated RAW 264.7 macrophages. Two independent experiments were carried out in triplicate and the results are presented as the mean ± standard deviation.

### 2.6. UPLC-QTOF-MS Analysis

The most active fractions were subjected to a fingerprint analysis by UPLC-UV-HRMS using a Xevo G2-XS QTOF spectrometer (Waters, Milford, MA, USA) coupled with an Acquity LC system equipped with MassLynx version 4.1 software (Waters, Milford, MA, USA). For analysis, 5 µL of each sample was placed on a HSS T3 column (100 mm × 2.10 mm, 1.8 µm, Waters, Milford, MA, USA). The mobile phase consisted of H_2_O + 0.1% formic acid (A), and acetonitrile (ACN) + 0.1% formic acid (B), and the gradient was set as follows (min/B%): 0.0/3, 1.0/3, 17.0/100, 19.0/100, 21.0/3 and 25.0/3. Full scan data were recorded in ESI (−) and ESI (+) MSe modes from *m/z* 50 to 1500. UV detection was performed at 210 and 254 nm. The conditions used for the ESI source were as follows: capillary voltage, 1.0 KV in ESI+, 0.8 kV in ESI-; sampling cone, 40 V; source temperature, 120 °C; desolvation temperature, 550 °C; cone and desolvation gas flow rates, 50 L/h and 1000 L/h, respectively. Nitrogen was used as cone and desolvation gas.

### 2.7. Data Processing

The UPLC-MS raw data were converted to abf files (Reifycs Abf Converter) and processed with MS-DIAL version 4.90 (https://doi.org/10.1038/nmeth.3393/ accessed on 16 July 2022) for mass signal extraction between 50 and 1500 Da from 0 to 25 min [25]. The MS1 and MS2 tolerance levels were set to 0.05 in the centroid mode. The optimized detection threshold was set to 8000 for MS1 and 5000 for MS2. Then, MS-FINDER software version 3.52 (https://doi.org/10.1021/acs.analchem.6b00770/ accessed on 16 July 2022) was used for in silico fragmentation predictions. Compounds were tentatively identified according to their similarity score, which was based on a comparison between experimental MS/MS fragments and in silico spectra. The MS1 and MS2 tolerances were set to 0.01 and 0.25 Da, respectively. The natural product databases integrated into MS-FINDER (ChEBI, PlantCyc, UNPD, NPA, KNApSAcK, PubChem, COCONUT and NANPDB) were used for compound identification and to predict the molecular structures of the compounds. The following public databases were also consulted: MassBank of North America (MoNA) (http://mona.fiehnlab.ucdavis.edu/ accessed on 16 July 2022), and NIST Mass Spectrometry Data Center (http://chemdata.nist.gov/).

### 2.8. Statistical Analysis

One-way analysis of variance (ANOVA) was used to test for differences in mean values between different samples, followed by Tukey’s honestly significant differences (HSD) multiple rank test at *p* < 0.05. To test for the inhibition of nitric oxide release, a Mann–Whitney test was used. All analyses were performed using Statgraphic^®^ Centurion XV for Windows version 15.2.14 (Stat Point, Inc., Warrenton, VA, USA) software.

## 3. Results

### 3.1. Free Radical Scavenging Activity

The free radical scavenging activity was determined using the common DPPH and ABTS methods. Table 1 shows the mean inhibitory concentration (IC_50_) of the samples in these radical scavenging assays. The results show that the *C. linearis* crude extract (**CEL**) had good antioxidant activity against both radicals: ABTS and DPPH, but especially against the ABTS radical with an IC_50_ value lower (IC_50_ 105.1 ± 7.43 μg/mL) than the positive control ascorbic acid (IC_50_ 182.35 ± 8.07 μg/mL). The dichloromethane (**DCM**) and ethyl acetate (**EAF**) fractions showed lower IC_50_ values than the crude extract (**CEL**). Once again, the ABTS radical was the most susceptible. In all cases, but especially for the CEL, DCM and EAF fractions, marked concentration-dependent behavior was observed, allowing for the estimation of the IC_50_ values in equations with R-square values over 0.75. In contrast, the *n*-hexane fraction (HF) demonstrated poor activity, with IC_50_ values of 406.22 and 426.74 μg/mL for the ABTS and DPPH, respectively, while the *n*-butanol fraction (BF) was not active at the tested concentrations. Similar results were observed for the DPPH radical (Table 1).

### 3.2. Cell Viability Assay

The effect of the samples on the RAW 264.7 cell viability was measured in a concentration range between 8 and 256 µg/mL. The results are shown in Table 1, highlighting that for CEL and HF, noteworthy cytotoxicity was observed at concentrations above 64 µg/mL, with CC_50_ values estimated as 75.30 and 70.12 µg/mL for CEL and HF, respectively. On the other hand, the DCM, EAF and BF fractions did not affect the cell viability at the tested doses (Figure 1). Based on those results, exploration of the inhibitory effect on nitric oxide release in LPS-stimulated macrophages of the extract/fractions was performed at concentrations below 64 µg/mL.

### 3.3. Inhibitory Effect on Nitric Oxide Release in LPS-Stimulated Macrophages

The effect of the crude extract and its fractions on NO release in LPS-stimulated RAW 264.7 macrophages was also assessed (Table 1). A significant inhibitory effect on nitric oxide release was detected for the *n*-hexane fraction (HF), followed by the crude extract (CEL) and the ethyl acetate fraction (EAF). The high activity observed for these three samples can be better appreciated in Figure 2 when compared with the nitric oxide inhibition release shown by the positive control L-NAME (100 µM). In all cases, a concentration-dependent relationship was observed. The other samples (DCM and BF) were not active or less active than the positive control (L-NAME) at the concentrations tested.

### 3.4. UPLC-QTOF-MS Analysis

In order to determine the metabolites present in the most active and non-toxic fractions (DCM and EAF), an UPLC-QTOF-MS analysis in the ESI (−) and ESI (+) modes was performed. The chromatograms obtained in both modes for both fractions are represented in Appendix A. From the 23 peaks detected, 18 compounds were tentatively identified, as is shown in Table 2. As can be seen in Table 2, compounds **5**, **6**, **9**, **20**, **21**, **22** and **23** are common to both fractions, while compounds **1**, **2**, **7** and **19** appear only in the DCM fraction. Of them, 13 include a nitrogen atom in their structure, thus belonging to the alkaloid family, while the other 10 are polyphenols. Within the alkaloid types, compounds **2**, **3**, **5** and **7** belong to the morphinane family, compounds **4** and **19** to the aporphine family and compounds **6**, **9** and **12** to the benzylisoquinoline family. A fragmentation proposal for two of the alkaloids is presented in Figure 3 and Figure 4. Meanwhile, within the polyphenols, compounds **11** and **13** are O-glucoronide-pentosyl derivatives of quercetin, **15**, **16** and **20** are O-deoxyhexo-syl-hexoside or O-p-coumaroyl hexoside derivates of kaempferol and compounds **17**, **18**, **21** and **22** are O-glucoside or O-p-coumaroyl hexoside derivatives of isorhamnetin. A fragmentation proposal for one of the flavonoid derivatives is presented in Figure 5.

## 4. Discussion

*Croton linearis* Jacq. is a small shrub endemic to the Bahamas and distributed along the southern coast of the eastern region of Cuba. Its ethnopharmacological use, treating inflammatory and infective diseases, as well as studies reporting the production of alkaloids and polyphenols, led us to explore its inhibitory effect on nitric oxide release in LPS-stimulated RAW 264.7 macrophages. Additionally, the free radical scavenging activity of the *Croton linearis* crude extract and its fractions was explored as a probable mechanism of action associated with some of the reported folk uses. With this purpose, the crude extract (CLE) was partitioned into four fractions, of which the active and RAW 264.7 non-toxic ones (DCM and EAF) were phytochemically characterized by means of UPLC-QTOF-MS analysis, allowing us to tentatively identify 18 compounds: nine alkaloids and nine flavonoid derivatives. Compounds such as laudanidine (peak 9 in Table 2), glaucine (peak 19 in Table 2) and isorhamnetin-*O-p*-coumaroyl hexoside (peak 21/22 in Table 2) were among those already reported in previous work [15]. Beyond those, we detected compounds belonging to three different alkaloid families (morphinane, aporphine and benzylisoquinoline), and several flavonoid glycosides from three different flavonol aglycones (quercetin, kaempferol and isorhamnetin). Compounds belonging to these three alkaloid families have already been reported in *C. linearis* [15,29], and derivatives of quercetin, kaempferol and isorhamnetin have already been reported, as well [15,17].

Overproduction of reactive nitrogen species (RNS) has been associated with oxidative stress, inflammatory conditions, immune response and other processes [30,31]. Recently, considerable attention has been focused on the use of antioxidants, especially natural antioxidants, to inhibit NO production [32]. NO is a relatively unreactive RNS, but one of its derivatives, peroxynitrite (ONOO^−^), is a powerful oxidant, capable of damaging many biological molecules [30]. NO, meanwhile, acts as a pro-inflammatory mediator associated with acute and chronic inflammation and the release of other pro-inflammatory mediators, such as bradykinins, and histamine [33,34].

The crude extract and the *n*-hexane fraction were the most active inhibitors of NO production in RAW 264.7 macrophages with IC_50_ values of 21.59 ± 2.97 and 4.88 ± 2.64 µg/mL, respectively; however, they also showed a noteworthy cytotoxic effect at concentrations over 64 µg/mL (Figure 3), limiting their possible future application. On the contrary, EAF exhibited a good inhibitory effect (IC_50_ 40.03 ± 2.94 µg/mL) on NO production in a concentration-dependent way (R^2^ 0.95 ± 0.01) without affecting cell viability, thus emerging as the most promising fraction.

With regard to the phytochemical analysis performed, the EAF fraction contained most of the flavonoids detected, especially those derived from quercetin and kaempferol. Quercetin and kaempferol, and their derivatives, have been reported as important inhibitors of NO release, even in macrophage models [35,36]. Several mechanisms have been reported, with prevalence of the direct inhibition of the inducible nitric oxide synthase (iNOS), cyclooxygenase-2 and reactive C-protein [37], as well as the downregulation of the NF-κB, STAT-1 and MAPK pathways [36]. On the other hand, the DCM fraction contained relatively more alkaloids, and to a lesser extent, flavonoids of the isorhamnetin-*O-p*-coumaroyl type. Although isorhamnetin has also been reported as an inhibitor of NO release, some studies indicate that it has lower activity compared to quercetin and kaempferol. These observations could explain the higher activity of the EAF fraction compared to the DCM fraction.

The effect on the NO production of extracts and compounds from other species of the genus *Croton* has been previously studied. Njoya et al., in 2018, reported the ethanolic root extract of *C. zambesicus* (*C. gratissimus*) and the acetone extract of *C. pseudopulchellus* to be good inhibitors of the NO release in LPS-stimulated RAW 264.7 macrophages, with IC_50_ values of 49.24 ± 0.93 μg/mL and 34.64 ± 0.06 μg/mL, respectively [38]. Recently, the potential anti-inflammatory properties of four South African *Croton* species—*C. gratissimus, C. pseudopulchellus*, *C. sylvaticus* and *C. steenkampianus*—was reported. These species demonstrated a high capacity to attenuate NO production with negligible cytotoxicity [39]. Associated with this activity, the phenylpropanoid derivatives obtained from *C. velutinus*, exhibited concentration-dependent suppressive activity on the production of nitrite and IL-1β by macrophages stimulated with LPS and IFN-γ [40].

Independent of the aforementioned mechanisms, the antioxidant capacity stands out as the most common characteristic associated with inhibition of NO release [41,42]. Multiple chemical mechanisms have been proposed for food antioxidants, with phenolic compounds noted such as flavonoids, tannins and phenolic acids [43,44]. In the same way, the presence of those phenolic compounds in extracts and fractions of *Croton* species has been correlated with the antioxidant potential [45,46]. In this regard, both EAF and DCM fractions, as well as the crude extract CEL, showed good antioxidant activity, sometimes with better IC_50_ values than the positive control, ascorbic acid. However, other metabolites with scavenging capacity, such as alkaloids, may be present, as well. Novello et al., in 2016, reported an indole alkaloid isolated from *C. echioides* with high antioxidant activity (IC_50_ 30.0 ± 0.7 µM) against the DPPH radical [47]. In the present research, we have described the presence of several alkaloids with free phenolic groups, as well as different kinds of flavonoids, which could be responsible for the antioxidant activity observed for the crude extract and its most active fractions (DCM and EAF).

Oxidative stress is implicated in various inflammatory conditions, with ROS and RNS responsible for part of that involvement [30,48]. In that sense, this study revealed the good scavenging capacity of the crude extract and its fractions on ABTS and DPPH radicals; at the same time, it showed strong inhibition of NO production in LPS-stimulated RAW 264.7 macrophages, which was superior when compared to L-NAME (100 µM). This antioxidant capacity, at least in part, suggests the potential of this species to reduce pro-inflammatory molecules generated during inflammation. Several studies have demonstrated the correlation between the antioxidant capacity and anti-inflammatory activity of extracts and compounds from plants [49,50,51]. Accordingly, they can be considered a good source of antioxidants, which might be beneficial for combating diseases related to oxidative damage. Nevertheless, other antioxidant and anti-inflammatory assays of these extracts must be performed to establish possible mechanisms of action.

## 5. Conclusions

The effect on nitric oxide release in LPS-stimulated macrophages and free radical scavenging activity of the well-characterized crude extract of *Croton linearis* and its fractions were reported here for the first time. CEL, DCM and EAF fractions were the most active as free radical scavengers, while CEL, HF and EAF fractions showed a strong effect on nitric oxide release in LPS-stimulated RAW 264.7 macrophages. However, CEL and HF showed noteworthy cytotoxicity at concentrations above 64 µg/mL. UPLC-QTOF-MS identification demonstrated the rich arsenal of potentially bioactive components in the DCM and EAF fractions, consisting mainly of alkaloids and flavonoids, which could be responsible for the good antioxidant effects observed. These findings reveal the natural antioxidant potential of this species, as well as its probable anti-inflammatory effect, an activity for which it is used in folk medicine, but with further attention required due to the cytotoxicity observed in some of its fractions. Additional studies will be essential to establish its possible mechanisms of action, safeness and its relationship with the constituents of *Croton linearis* leaves.

## Figures and Tables

**Figure 1 antioxidants-11-01915-f001:**
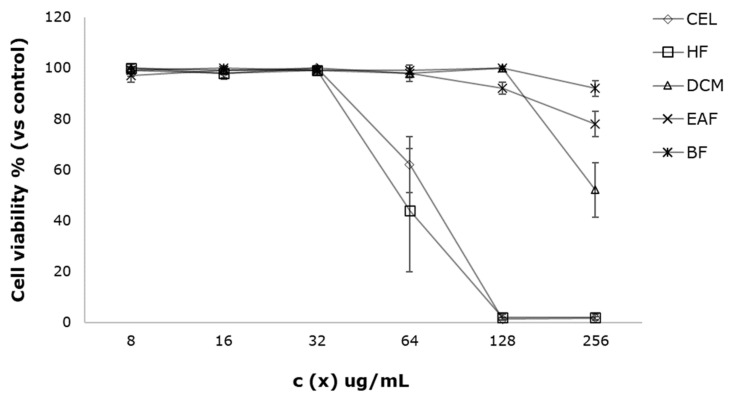
Effect of crude extract (CEL), *n*-hexane fraction (HF), dichloromethane fraction (DCM), ethyl acetate fraction (EAF) and *n*-butanol fraction (BF) on RAW 264.7 murine macrophage cell viability. The effect on the cell viability was determined by the resazurin assay. All values are expressed as the mean ± SD of three replicates. Tamoxifen was the reference control drug for cytotoxicity.

**Figure 2 antioxidants-11-01915-f002:**
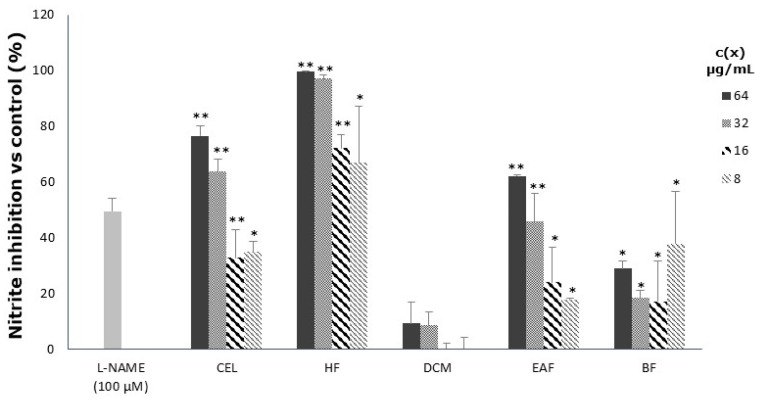
Effect of crude extract (CEL), n-hexane fraction (HF), dichloromethane fraction (DCM), ethyl acetate fraction (EAF) and n-butanol fraction (BF) on nitric oxide production in LPS-stimulated RAW 264.7 macrophages. The extracellular nitrite accumulation was measured by the Griess reaction. * and ** mean significant statistical differences with the untreated group at (*p* < 0.05) and (*p* < 0.01), respectively, according to a Mann–Whitney test.

**Figure 3 antioxidants-11-01915-f003:**
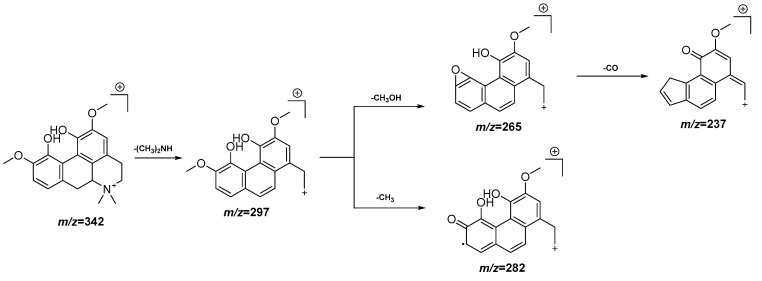
Peak 4, tentative identification of magnoflorine in the EAF fraction of *Croton linearis* Jacq. leaves. Hypothetical fragmentation pattern adapted from Jiao et al. [26].

**Figure 4 antioxidants-11-01915-f004:**
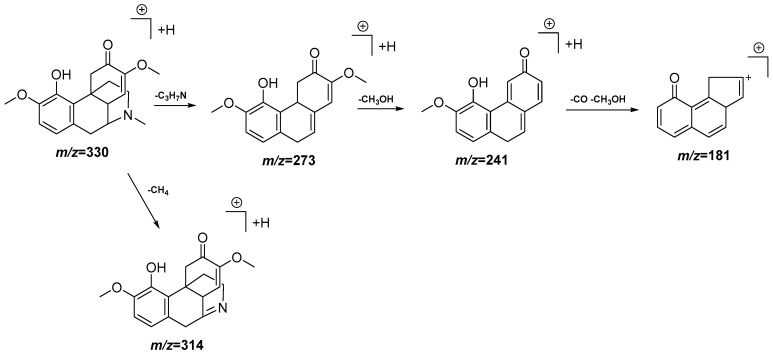
Peak 7, tentative identification of sinomenine in the DCM fraction of *Croton linearis* Jacq. leaves. Hypothetical fragmentation pattern adapted from Jiang et al. [27].

**Figure 5 antioxidants-11-01915-f005:**
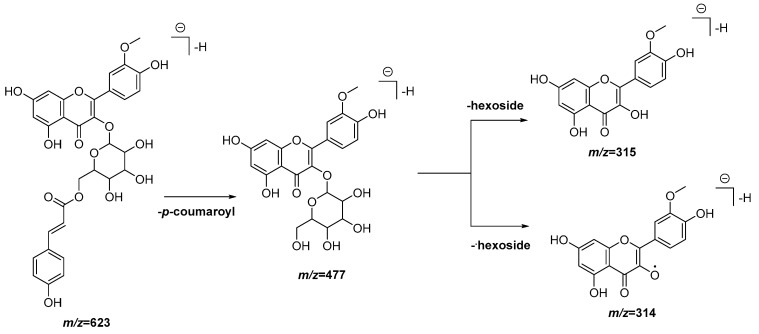
Peak 21, tentative identification of isorhamnetin-*O-p*-coumaroyl hexoside 1 (position of the substituents is arbitrary) in the DCM and EAF fractions of *Croton linearis* Jacq. leaves. Hypothetical fragmentation pattern adapted from Panighel et al., 2015 [28].

**Table 1 antioxidants-11-01915-t001:** DPPH and ABTS radical scavenging activity, NO inhibitory effect and cell viability of *C. linearis* leaves’ extract and its fractions.

Sample	ABTS IC_50_ (µg/mL)	DPPH IC_50_ (µg/mL)	Nitric Oxide Inhibitory Activity IC_50_ (µg/mL)	Cytotoxicity in RAW 264.7 Cells CC_50_ (µg/mL)
CEL	105.10 ± 7.43 ^b^	153.76 ± 23.61 ^c^	21.59 ± 2.97 ^b^	75.30 ± 3.71 ^a^
HF	406.22 ± 25.82 ^d^	426.74 ± 22.00 ^d^	4.88 ± 2.64 ^a^	70.12 ± 7.21 ^a^
DCM	55.93 ± 10.75 ^a^	13.47 ± 3.17 ^a^	>64	>256
EAF	35.53 ± 3.33 ^a^	84.23 ± 6.95 ^b^	40.03 ± 2.94 ^c^	>256
BF	>500	>500	>64	>256
AA	182.35 ± 8.07 ^c^	28.49 ± 2.91 ^a^	ND	ND
Tamoxifen	ND	ND	ND	1.41 ± 0.21

Note: Data are expressed as the mean ± standard deviation. Crude extract (**CEL**), *n*-hexane fraction (**HF**), dichloromethane fraction (**DCM**), ethyl acetate fraction (**EAF**), *n*-butanol fraction (**BF**) and ascorbic acid (**AA**). ND: not determined. Different superscripts within the same column indicate significant differences between extracts/fractions (*p* < 0.05) according to a one-way ANOVA test followed by Tukey’s test (n = 3).

**Table 2 antioxidants-11-01915-t002:** Assigned compounds, molecular ions in the ESI (−) or ESI (+) modes and fragment ions of the main components present in the dichloromethane (DCM) and ethyl acetate (EAF) fractions of *Croton linearis* Jacq. leaves.

Peak No.	Rt (min)	Precursor Ion (*m/z*)	Theoretical Mass (*m/z*)	Accuracy (ppm)	MS/MS Ions	Precursor Ion Formula	Tentative Identification	Extract
1	3.09	334.1655 [M + H]^+^	334.1654	0.3	318.1709, 190.6034	C_18_H_23_NO_5_	Unknown	DCM
2	3.16	318.1707 [M + H]^+^	318.1705	0.6	243.1026, 211.0762, 181.0654	C_18_H_23_NO_4_	*N*-demethyl dihydrosinomenine	DCM
3	3.35	334.1656 [M + H]^+^	334.1654	0.6	259.0967, 213.0911, 188.0712	C_18_H_23_NO_5_	Sinococuline	EAF
4	3.73	342.1703 [M]^+^	342.1705	−0.6	297.1128, 282.0896, 265.0868, 237.0917	C_20_H_24_NO_4_^+^	Magnoflorine	EAF
5	3.84	332.1499 [M + H]^+^	332.1498	0.1	275.0918, 227.0708	C_18_H_21_NO_5_	*N*-oxide demethyl sinomenine	EAF, DCM
6	4.04	476.1945 [M + H]^+^	476.1921	5.0	326.1244, 314.1393, 180.0660, 164.0711	C_24_H_29_NO_9_	Dauricoside (1)	EAF, DCM
7	4.12	330.1717 [M + H]^+^	330.1705	3.6	314.1402, 273.1138, 241.0874, 181.0657	C_19_H_23_NO_4_	Sinomenine	DCM
8	4.24	760.2825 [M + H]^+^	760.2817	1.1	638.2453, 598.2291, 565.1902, 326.1239, 164.0710	C_37_H_45_NO_16_	Unknown	EAF
9	4.38	344.1862 [M + H]^+^	344.1862	0.0	299.1291, 267.1029, 192.1027, 175.0765	C_20_H_25_NO_4_	Laudanidine	EAF, DCM
10	4.45	612.2462 [M + H]^+^	612.2445	2.8	478.2087, 344.1858, 342.1710, 178.0870	C_32_H_37_NO_11_	Unknown	EAF
11	4.68	741.1842 [M − H]^−^	741.1878	−4.9	609.1470, 477.1773, 301.0334, 300.0270	C_32_H_38_O_20_	Quercetin-*O*-pentosyl-pentosyl- glucuronide	EAF
12	4.78	476.1929 [M + H]^+^	476.1921	1.7	358.2010, 314.1391, 175.0757	C_24_H_29_NO_9_	Dauricoside (2)	EAF
13	4.99	609.1456 [M − H]^−^	609.1456	0.0	301.0351, 300.0281	C_27_H_30_O_16_	Quercetin- *O*- pentosyl- glucuronide	EAF
14	5.18	952.3251	952.3239	1.3	462.1924, 303.0508	C_47_H_53_NO_20_	Unknown	EAF
15	5.26	593.1503 [M − H]^−^	593.1506	−0.5	285.0394 284.0324	C_27_H_30_O_15_	Kaempferol-*O*- deoxyhexosyl-hexoside (1)	EAF
16	5.38	593.1500 [M − H]^−^	593.1506	−1.0	285.0399 284.0326	C_27_H_30_O_15_	Kaempferol- *O*- deoxyhexosyl-hexoside (2)	EAF
17	5.60	477.1031 [M − H]^−^	477.1033	−0.4	447.0953, 315.0496, 314.0439, 285.0401, 284.0331, 271.0251, 255.0298, 243.0297,	C_22_H_22_O_12_	Isorhamnetin 3-*O*-glucoside	EAF
18	5.68	477.1031 [M − H]^−^	477.1038	1.0	315.0496, 314.0441, 285.0410, 271.0253, 243.0297	C_22_H_22_O_12_	Isorhamnetin 7-*O*-glucoside	EAF
19	5.95	356.1870 [M + H]^+^	356.1862	2.2	325.1547, 310.1197, 294.1272	C_21_H_25_NO_4_	Glaucine	DCM
20	6.71	593.1284 [M − H]^−^	593.1295	−1.9	447.0956, 285.0406, 284.0329	C_30_H_26_O_13_	Tiliroside	EAF, DCM
21	6.84	623.1404 [M − H]^−^	623.1401	0.5	477.1058, 315.0513, 314.0438	C_31_H_28_O_14_	Isorhamnetin- *O-p*-coumaroyl hexoside (1)	EAF, DCM
22	6.90	623.1393 [M − H]^−^	623.1401	−1.3	477.1062, 315.0517, 314.0440	C_31_H_28_O_14_	Isorhamnetin-*O-p*-coumaroyl hexoside (2)	EAF, DCM
23	7.40	880.4304 [M + Na]^+^ 858.4481 [M + H]^+^	858.4487	−1.3	429.7295	C_42_H_67_NO_17_	Unknown	EAF, DCM

## Data Availability

Not applicable.

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
