# Peer review of "Inhibitory Effect on Nitric Oxide Release in LPS-Stimulated Macrophages and Free Radical Scavenging Activity of Croton linearis Jacq. Leaves"

_antioxidants, 2022, doi:10.3390/antiox11101915_

Round 1

Reviewer 1 Report

The authors have extensive experience in the discussed scientific field and in the proposed research methodology. The methodological description of the obtained results, and in particular the discussion of obtained results and possible causes, were very interesting.

Below, I’ve presented some minor editorial comments, which, in the opinion of the reviewer, will facilitate the reading of the text of the manuscript for the reader.

2.1. Solvents and reagents

1. I propose to include information about the source of RAW 264.7 macrophages.

2.2. Samples

1. Line 97: I propose to supplement the first sentence with information on whether the obtained samples came from one site, or from many (which?)? - then the reader would not have to refer to the quoted source works. Not only that, one might mistakenly get the impression that the samples obtained for analysis came from the previous, cited studies - and they probably are not….

2. Line 97: maybe it should be considered replacing the word “researches” with 'studies”/ “investigations” / “analyzes”, or something like that ?.

2.5. Cell Viability Assay

1. Line 157: Please explain the abbreviation FCS here. 

2. Line 162: I suggest replacing the word "are" with "were".

2.6. UPLC-QTOF-MS analysis

1. Line 172: Please explain the abbreviation ACN here. 

3. Results

3.1. Free radical scavenging activity

1. In the legend of Table 1, in the description of statistical significance - in the description of the meaning of the symbols: a - d, there is mentioned statistical significance in each of the columns, but it is not explicitly mentioned for which control group (i.e. whether for Tamoxifen or vs the appropriate study group?) . I kindly ask you to clarify this information.

3.2. Inhibitory effect on nitric oxide release in LPS-stimulated macrophages

1. In lines: 225 and 230 - I propose to use the past tense.

3.3. Cell Viability Assay

1. Line: 243 - Here, again, I kindly propose to use the past tense. 

3.4. UPLC-QTOF-MS analysis

1. In the description of the legend of Figure 4, in lines: 273, 277, 281 - I’d consider removing the dates from the quoted authors, especially since the number of the source publication is given just after. But that's just a suggestion to think about.

4. Discussion

1. In the first part of this section of the manuscript, I missed one sentence clearly informing the reader why the CEL, as well as the other two fractions, were not subjected to the above-discussed preliminary phytochemical analysis. This is important because both the CEL and the HF fraction showed quite good antioxidant activities, which was also mentioned in line 342. Does it result from the finding of the significant cytotoxic effect at concentrations over 32 µg/mL for them? This is only a comment, a suggestion for consideration for adding.

Author Response

2.1. Solvents and reagents

  1. I propose to include information about the source of RAW 264.7 macrophages.

R/ Done. Thanks

2.2. Samples

  1. Line 97: I propose to supplement the first sentence with information on whether the obtained samples came from one site, or from many (which?)? - then the reader would not have to refer to the quoted source works. Not only that, one might mistakenly get the impression that the samples obtained for analysis came from the previous, cited studies - and they probably are not….

R/ Done. Thanks

  1. Line 97: maybe it should be considered replacing the word “researches” with 'studies”/ “investigations” / “analyzes”, or something like that ?.

 R/ Done. Thanks

2.5. Cell Viability Assay

  1. Line 157: Please explain the abbreviation FCS here. 

 R/ Done. Thanks

  1. Line 162: I suggest replacing the word "are" with "were".

R/ Done. Thanks

2.6. UPLC-QTOF-MS analysis

  1. Line 172: Please explain the abbreviation ACN here. 

R/ Done. Thanks

  1. Results

3.1. Free radical scavenging activity

  1. In the legend of Table 1, in the description of statistical significance - in the description of the meaning of the symbols: a - d, there is mentioned statistical significance in each of the columns, but it is not explicitly mentioned for which control group (i.e. whether for Tamoxifen or vs the appropriate study group?) . I kindly ask you to clarify this information.

R/ Added comments in table legend for clarity.

3.2. Inhibitory effect on nitric oxide release in LPS-stimulated macrophages

  1. In lines: 225 and 230 - I propose to use the past tense.

 R/ Modifications were done.

3.3. Cell Viability Assay

  1. Line: 243 - Here, again, I kindly propose to use the past tense. 

  R/ Modifications were done. Thanks

3.4. UPLC-QTOF-MS analysis

  1. In the description of the legend of Figure 4, in lines: 273, 277, 281 - I’d consider removing the dates from the quoted authors, especially since the number of the source publication is given just after. But that's just a suggestion to think about.

 R/ Done. Thanks

  1. Discussion
  2. In the first part of this section of the manuscript, I missed one sentence clearly informing the reader why the CEL, as well as the other two fractions, were not subjected to the above-discussed preliminary phytochemical analysis. This is important because both the CEL and the HF fraction showed quite good antioxidant activities, which was also mentioned in line 342. Does it result from the finding of the significant cytotoxic effect at concentrations over 32 µg/mL for them? This is only a comment, a suggestion for consideration for adding.

R/ Small comments to clarify this part were added to paragraphs 1 and 3 of the discussion.

Reviewer 2 Report

In this manuscript, the authors report a detailed experimental investigation on the effect on nitric oxide release in LPS-stimulated macrophages and free radical scavenging activity of the well characterize crude extract of Croton linearis and its fractions. The work has been competently done, results are well presented and the conclusions sound good. I suggest the publication after the consideration on the following minor points:

-the bibliography is not accurate. The following article on the antioxidant working mechanisms must be included: Annu. Rev. Food Sci. Technol. 2016, 7, 335–352; Food Chem. 2011, 125, 288–306: Phys. Chem. Chem. Phys., 2013, 15, 4642-4650;

-some typos must be removed.

Author Response

In this manuscript, the authors report a detailed experimental investigation on the effect on nitric oxide release in LPS-stimulated macrophages and free radical scavenging activity of the well characterize crude extract of Croton linearis and its fractions. The work has been competently done, results are well presented and the conclusions sound good. I suggest the publication after the consideration on the following minor points:

-the bibliography is not accurate. The following article on the antioxidant working mechanisms must be included: Annu. Rev. Food Sci. Technol. 2016, 7, 335–352; Food Chem. 2011, 125, 288–306: Phys. Chem. Chem. Phys., 2013, 15, 4642-4650;

-some typos must be removed.

R/ A sentence including two of the mentioned references was added.

  • Rev. Food Sci. Technol. 2016, 7, 335–352;
  • Food Chem. 2011, 125, 288–306:

We do not include the third paper suggested by you because we don’t use TROLOX as model/positive control. Thanks for your observations.

Reviewer 3 Report

The reviewed manuscript concerns the antiinflammatory potential of different fractions form Croton linearis. The manuscript is written in good English and the experiments is designed appropiate, with one exception: I am very concerned on the cytotoxic effect of some of the tested samples to RAW macrophages. In my opinion, the antiinflammatory assay should not be performed at the doses, at which the cytotoxic impact was observed. Thus, I recommend to present the results only for the doses up to 32 μg/mL, or to present the results for all the doses but without the two mentioned fractions with the cytotoxic effect. I cannot agree with the authors that the cytotoxic effect was moderate - in my opinion it was quite high. Moreover, I would suggest to change the layouts for figures 1 and 2, as for me they are not clear enough. As far as the conclusions are concerned, some issues should be raised referring to the safety of the extract and the fractions in their potential future use.

Author Response

The reviewed manuscript concerns the antiinflammatory potential of different fractions form Croton linearis. The manuscript is written in good English and the experiments is designed appropiate, with one exception: I am very concerned on the cytotoxic effect of some of the tested samples to RAW macrophages. In my opinion, the antiinflammatory assay should not be performed at the doses, at which the cytotoxic impact was observed. Thus, I recommend to present the results only for the doses up to 32 μg/mL, or to present the results for all the doses but without the two mentioned fractions with the cytotoxic effect. I cannot agree with the authors that the cytotoxic effect was moderate - in my opinion it was quite high. Moreover, I would suggest to change the layouts for figures 1 and 2, as for me they are not clear enough. As far as the conclusions are concerned, some issues should be raised referring to the safety of the extract and the fractions in their potential future use.

R/

  • The concentrations at which the toxic effect was observed were removed (see figure 2). Now the NO release inhibition is presented just for the concentrations bellow the CC50
  • The “moderate cytotoxic effect” expression was changed for “noteworthy cytotoxic effect” in the entire document, but to be consistent with the CC50 calculated we also change the concentration limit for such expression from 32 up to 64 μg/mL.
  • The topics dealing with “Cytotoxicity” and “NO release” were changed in order of appearance, to illustrate clearly the decision to skip those toxic concentrations in the NO release test. The design of the figure related to “NO release” was changed in order to make more clearly the results obtained.
  • Conclusions were modified including some comments regarding the cytotoxic effect of some of the fraction and in consequence calling the attention regarding their safety.
  • Thanks for all the observations.